# Fault-tolerant interface between quantum memories and quantum processors

Hendrik Poulsen Nautrup [1], Nicolai Friis [1,2] & Hans J. Briegel[1]

Topological error correction codes are promising candidates to protect quantum computations from the deteriorating effects of noise. While some codes provide high noise thresholds suitable for robust quantum memories, others allow straightforward gate implementation needed for data processing. To exploit the particular advantages of different topological codes for fault-tolerant quantum computation, it is necessary to be able to switch between them. Here we propose a practical solution, subsystem lattice surgery, which requires only two-body nearest-neighbor interactions in a fixed layout in addition to the indispensable error correction. This method can be used for the fault-tolerant transfer of quantum information between arbitrary topological subsystem codes in two dimensions and beyond. In particular, it can be employed to create a simple interface, a quantum bus, between noise resilient surface code memories and flexible color code processors.

[1] Institute for Theoretical Physics, University of Innsbruck, Technikerstr. 21a, 6020 Innsbruck, Austria. [2] Institute for Quantum Optics and Quantum Information, Austrian Academy of Sciences, Boltzmanngasse 3, 1090 Vienna, Austria. Correspondence and requests for materials should be addressed to H.P.N. (email: hendrik.poulsen-nautrup@uibk.ac.at)

Noise and decoherence can be considered as the major obstacles for large-scale quantum information processing. These problems can be overcome by fault-tolerant quantum computation[1,2], which holds the promise of protecting a quantum computer from decoherence for arbitrarily long times, provided the noise is below a certain threshold. Quantum error correction codes are indispensable for any fault-tolerant quantum computation scheme[3]. Among these, stabilizer codes[4], building on classical coding theory, admit a particularly compact description. In particular, the subclass of topological stabilizer codes (TSCs)[5] is promising since TSCs are scalable and permit a local description on regular $D$-dimensional lattices.

Two of the most prominent examples of TSCs in two dimensions are surface codes[6,7] and color codes[8]. While surface codes support adaption to biased noise[9] and have better error thresholds than comparable color codes[10], they do not support a transversal phase gate. However, transversality is essential for any encoded quantum logic gate since it guarantees a straightforward implementation within a quantum error correction code. Gauge color codes, for example, were recently shown to support the transversal implementation of a universal set of gates in a three-dimensional (3D) layout[11]. It is hence desirable to store quantum information in surface code quantum memories while performing computation on color codes in two (and three) dimensions[12].

Here we present a protocol that makes such hybrid computational architectures viable. We develop a simple, fault-tolerant conversion scheme between two-dimensional (2D) surface and color codes of arbitrary size. Our flexible algorithm for code switching is based on a formalization of lattice surgery[13,14] in terms of operator quantum error correction[15] and measurement-based quantum computation[16]. This introduces the notion of subsystem lattice surgery (SLS), a procedure that can be understood as initializing and gauge fixing a single subsystem code. The required operations act locally on the boundaries, preserve the 2D structure, and are generators of a topological code. As all generators of the resulting code have constant size, errors on any of their components only affect a constant number of qubits, making the protocol inherently fault-tolerant. As we explicitly show, this generalizes the methodology of lattice surgery to any combination of 2D topological subsystem stabilizer codes, with color-to-surface code switching as an example of particular interest. While we restrict ourselves to 2D topological codes for the better part of this paper, we show that the essential ingredients of SLS carry over to higher-dimensional codes. In fact, the procedure works even for non-topological codes at the expense of locality. Therefore, our results represent a significant step toward a fault-tolerant interface between robust quantum memories and versatile quantum processors, independently of which topological codes will ultimately prove to be most effective. The method proposed here hence has the prospect of connecting different components of a future quantum computer in an elegant, practical, and simple fashion.

## Results

**Stabilizer formalism.** We consider a system comprised of $n$ qubits with Hilbert space $\mathcal{H} = \left(\mathbb{C}^2\right)^{\otimes n}$. The group of Pauli operators $\mathcal{P}_n$ on $\mathcal{H}$ is generated under multiplication by $n$ independent Pauli operators and the imaginary unit $i$. We write $\mathcal{P}_n = \langle i, X_1, Z_1, ..., X_n, Z_n \rangle$, where $X_j, Z_j$ are single-qubit Pauli operators acting on qubit $j$. An element $P \in \mathcal{P}_n$ has weight $w(P)$ if it acts nontrivially on $w$ qubits. We define the stabilizer group $\mathcal{S} = \langle S_1, ..., S_s \rangle$ for $s \leq n$ as an Abelian subgroup of $\mathcal{P}_n$ such that the generators $S_i$ are independent operators $\forall i = 1, ..., s$ and $-\mathbb{1} \notin \mathcal{S}$. $\mathcal{S}$ defines a $2^k$-dimensional codespace $\mathcal{C} = \text{span}(\{|\psi\rangle\})$ of codewords $|\psi\rangle \in \mathcal{H}$ through the condition $S|\psi\rangle = |\psi\rangle \, \forall S \in \mathcal{S}$,

encoding $k = n - s$ qubits. We denote the normalizer of $\mathcal{S}$ by $N(\mathcal{S})$, which here is the subgroup of $\mathcal{P}_n$ that commutes with all elements of $\mathcal{S}$. That is, elements of $N(\mathcal{S})$ map the codespace to itself. We write $\mathcal{L} = N(\mathcal{S})/\mathcal{S}$ for the (quotient) group of logical operators which induces a tensor product structure on $\mathcal{C}$, i.e., $\mathcal{C} = \otimes_{i=1}^k \mathcal{H}_i$. This construction implies that different logical Pauli operators are distinct, non-trivial classes of operators with an equivalence relation given by multiplication of stabilizers.

The distance of an error correction code is the smallest weight of an error $E \in \mathcal{P}_n$ such that $E$ is undetectable. The code can correct any set of errors $\mathcal{E} = \{E_a\}_a$ iff $E_a E_b \notin N(\mathcal{S}) - \langle i \rangle \mathcal{S}$ $\forall E_a, E_b \in \mathcal{E}$. A stabilizer code defined through $\mathcal{S}$ has distance $d$ iff $N(\mathcal{S}) - \langle i \rangle \mathcal{S}$ contains no elements of weight less than $d$. Equivalently, any non-trivial element of $\mathcal{L}$ is supported on at least $d$ qubits. By the above error-correction condition, an error with weight at most $(d-1)/2$ can be corrected while an error $E$ with weight $d/2 \leq w(E) < d$ can only be detected. From a slightly different perspective, codewords are degenerate ground states of the Hamiltonian $H = -\sum_{i=1}^s S_i$. Adopting this viewpoint, correctable errors are local excitations in the eigenspace of $H$ since they anticommute with at least one generator $S_i \in \mathcal{S}$, while logical operators map the degenerate ground space of $H$ to itself.

**Subsystem stabilizer formalism.** A subsystem structure can be induced on stabilizer codes by considering non-Abelian gauge symmetries[15]. The group of gauge transformations is defined as $\mathcal{G} = \mathcal{L}_G \times \mathcal{S} \times \langle i \rangle$ where $\mathcal{S}$ is a stabilizer group and $\mathcal{L}_G$ is the group of operators in $N(\mathcal{S}) - \langle i \rangle \mathcal{S}$ that are not in a non-trivial class of $\mathcal{L} = N(\mathcal{S})/\mathcal{G}$. This imposes a subsystem structure on the codespace $\mathcal{C} = \mathcal{H}_L \otimes \mathcal{H}_G$ where all operators in $\mathcal{G}$ act trivially on the logical subspace $\mathcal{H}_L$[17]. While logical operators in $\mathcal{L}$ define logical qubits in $\mathcal{H}_L$, operators in $\mathcal{L}_G$ define so-called gauge qubits in $\mathcal{H}_G$. That is, operations in $\mathcal{L}$ and $\mathcal{L}_G$ both come in pairs of (encoded) Pauli $X$ and $Z$ operators for each logical and gauge qubit, respectively. We recover the stabilizer formalism if $\mathcal{G}$ is Abelian, i.e., $\mathcal{L}_G = \emptyset$. As before, a set of errors $\mathcal{E} = \{E_a\}_a$ is correctable iff $E_a E_b \notin N(\mathcal{S}) - \mathcal{G} \, \forall E_a, E_b \in \mathcal{E}$. Errors can again be considered as local excitations in the eigenspace of a Hamiltonian $H = -\sum_{i=1}^g G_i$, where $g$ is the number of generators $G_i \in \mathcal{G}$.

**Topological stabilizer codes.** Stabilizer codes $\mathcal{S}$ are called topological if generators $S_i \in \mathcal{S}$ have local support on a $D$-dimensional lattice. Here we focus on 2D rectangular lattices $\Lambda = [1, L] \times [1, L']$ with linear dimensions $L$ and $L'$ in the horizontal and vertical direction, respectively, where qubits are located on vertices (not necessarily all are occupied) but our discussion can be easily extended to arbitrary regular lattices. We call a generator $S_i \in \mathcal{S}$ local[5] if it has support within a square containing at most $r^2$ vertices for some constant $r$, called interaction range or diameter. Moreover, we require of a TSC that its distance $d$ can be made arbitrarily large by increasing the lattice size. In other words, the generators $S_i$ are not only local but also translationally invariant at a suitable scale[18]. This definition of TSCs can be extended straightforwardly to topological subsystem stabilizer codes (TSSCs)[19] where we impose locality on the generators of $\mathcal{G}$ instead of $\mathcal{S}$. Then, generators of $\mathcal{S}$ are not necessarily local.

**Logical string operators.** An intriguing feature of 2D topological codes is that logical operators can be interpreted as moving point-like excitations around the lattice, i.e. logical operators form strings across the lattice. Specifically, for any 2D TSC or TSSC on a lattice with open boundaries there exists a quasi-1D string of tensor products of Pauli operators generating a logical Pauli operator whose support can be covered by a rectangle of size $r \times L'$[5], i.e., a vertical strip of width at most $r$. If $\mathcal{S}$ obeys the

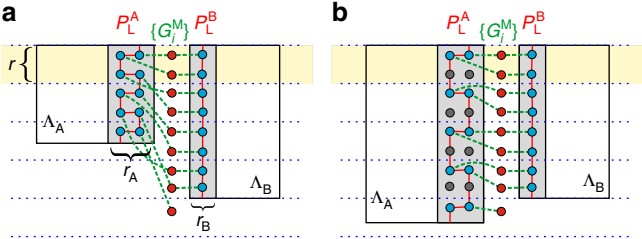

**Fig. 1** Subsystem lattice surgery between two topological codes. **a** Depiction of two topological codes defined on lattices $\Lambda_A$ and $\Lambda_B$ (grid not shown), respectively. Logical Pauli operators $P_L^l$ of the respective codes $l =$ A, B are represented by solid red lines connecting qubits (blue dots) within $\Lambda_A$ and $\Lambda_B$, respectively. These operators have support on a vertical strip (gray areas) of width $r_l$ for $l =$ A,B. Qubits in the support of $P_L^l$ are associated with a set $Q_l$. Ancillas added in step (i) of the merging protocol are depicted as red dots and are associated with a set $Q_C$. Merging operations $G_i^M$, which are defined in Eq. (2), are indicated (in part) by dashed green lines (with only the $i$th component being covered). The product of all merging operations acts as $P_L^A \otimes P_L^B$ such that a collective measurement projects onto a joint eigenstate of the two logical operators. Here, the diameter of an operator $G_i^M$ can grow with the lattice sizes since $r_A > r_B$. **b** Adding redundant vertices (gray dots) to the lattices increases the interaction range of the codes such that $Q_A$, $Q_B$, and $Q_C$ contain the same number of qubits within each horizontal strip of thickness $r$ between adjacent blue, dashed lines (e.g., the yellow area). Then, the diameter of merging operators can be kept constant, i.e., independent of the system size

locality condition, logical operators can be implemented row-by-row by applying Pauli operations along paths connecting two boundaries such that excitations emerge only at its endpoints. If $\mathcal{S}$ is not local, excitations might also emerge elsewhere since logical operators can anticommute with generators that act nontrivially on gauge qubits. Due to the translational invariance of generators with respect to the underlying lattice, one can always increase the code distance or choose the lattice such that there exists a logical Pauli operator whose support is covered by a vertical strip of maximal width $r$ along the boundary (or at most a constant number of lattice sites away from it).

Since it is a general feature of topological codes to support logical string operators[5], we can generalize the method of lattice surgery[13,14] to such codes. To this end, we consider any two TSCs or TSSCs $\mathcal{G}^A$ and $\mathcal{G}^B$ on two 2D lattices, e.g., $\Lambda_A = [1, L_A]^2$ and $\Lambda_B = [1, L_B]^2$, with open boundaries that have distances $d_A$ and $d_B$, respectively. We place the lattices such that their vertical boundaries are aligned. Let $P_L^A \in \mathcal{L}^A$ and $P_L^B \in \mathcal{L}^B$ be two logical operators defined along the aligned boundaries with maximal width $r_A$ and $r_B$, respectively, where $r_I$ is the interaction range of code $I =$ A, B. The logical operators act nontrivially on a set of qubits, say $Q_A$ and $Q_B$, respectively. Let $N_I = |Q_I|$ (for $I =$ A, B) and $N = \max\{N_A, N_B\}$. W.l.o.g. we assume $N_A = N$. We order the sets $Q_I$ according to walks along paths[5] on $\Lambda_I$ running, e.g., from top to bottom such that we can implement $P_L^I$ in a step-by-step fashion as described above. We write $Q_I = \{1, 2, ..., N_I\}$ for such an ordering. For ease of notation, we denote qubits in the support of $P_L^I$ by $i = 1, ..., N_I$ since the association of qubits to the sets $Q_I$ is clear from the context. Consequently, the logical operators take the form

$$P_L^I = P_{L,1}^I \otimes P_{L,2}^I \otimes ... \otimes P_{L,N_I}^I, \qquad (1)$$

for single-qubit Pauli operators $P_{L,i}^I$ acting on qubits $i$ with $i = 1, ..., N_I$.

**Merging protocol and code splitting.** Given the two codes $\mathcal{G}^A$ and $\mathcal{G}^B$ with respective logical Pauli operators $P_L^A$ and $P_L^B$ along boundaries, the goal is to achieve a mapping $\mathcal{G}^A \times \mathcal{G}^B \mapsto \mathcal{G}^A \times \mathcal{G}^B \times \langle P_L^A \otimes P_L^B \rangle$ that projects onto an eigenstate of $P_L^A \otimes P_L^B$ (i.e., a Bell state). Therefore, we now define a fault-tolerant protocol that merges the two codes into one, similar to the method of lattice surgery, which has been developed for surface codes[13] and extended to color codes[14]. As we will see, the merged code can then be split to realize the mapping described above. The merging procedure has the following four steps.

(i) Including ancillas. We introduce a set $Q_C$ of $N - 1$ ancillas initialized in the +1 eigenstates of their respective Pauli $X$ operators. The ancillas are placed on newly added vertices along a column between boundaries (Fig. 1a) and we order the set $Q_C$ accordingly from top to bottom, i.e., $Q_C = \{1, 2, ..., N - 1\}$. More formally, including ancillas in $|+\rangle$-states is equivalent to adding $N - 1$ single-qubit $X$ stabilizers to $\mathcal{G}^A \times \mathcal{G}^B$. Note that this step is not strictly necessary but it has been included here to highlight the similarity to lattice surgery. Nonetheless, this step is useful if one operates solely on TSCs (rather than TSSCs), since this can allow reducing the interaction range of the resulting merged code (see Methods section for details).

(ii) Preparing lattices. Redundant vertices are added to the lattices $\Lambda_A$ and $\Lambda_B$ without adding corresponding new qubits. This corresponds to a relabeling that increases the interaction range on both lattices and the distance between ancillas by at most a constant $r^2$, where $r = \max\{r_A, r_B\}$. This is done in such a way that horizontal strips of width (height) $r$ contain the same number of qubits in $Q_A$, $Q_B$, and $Q_C$ (Fig. 1b). Beyond the strip containing the last element of $Q_B$ we only require the same number of qubits in $Q_A$ and $Q_C$, except for the last strip, where $Q_C$ contains one qubit less than $Q_A$. The lattices are then connected along their vertical boundary such that the resulting merged lattice $\Lambda_M$ has linear dimension $L_A + L_B + 1$ along the horizontal direction. This step guarantees that the merged code remains topological according to the definition above. If a different definition of locality is adopted for topological codes, step (ii) can be replaced by a stretching of the lattices to ensure that the merged code is topological in the same sense as well.

(iii) Merging codes. After combining the underlying lattices, the codes are merged. To this end, one measures $N$ merging operators, which are defined on the new lattice $\Lambda_M$ as

$$G_i^M = P_{L,i}^A P_{L,i}^B Z_i^C Z_{i-1}^C \quad \forall i = 1, ..., N, \qquad (2)$$

where $Z_i^C$ acts on ancilla $i$ with $Z_0 \equiv \mathbb{1}$ and $Z_N \equiv \mathbb{1}$. Since $N_A \geq N_B$, we identify $P_{L,i}^B \equiv \mathbb{1}$ for $i > N_B$.

(iv) Correcting errors. In order to retain full fault tolerance, $d_{\min} = \min\{d_A, d_B\}$ rounds of error correction are required on the merged code. For this purpose, the structure of the merged code can be deduced from the SLS formalism that is introduced and analyzed in Methods section.

We now formulate the following theorem.

**Theorem 1.** For any two TSCs (or TSSCs) defined on regular 2D lattices with open boundaries, the procedure described in steps (i)–(iv) fault-tolerantly projects onto an eigenstate of $P_L^A P_L^B$. Moreover, all operations in steps (iii) and (iv) correspond to generators of a TSSC defined on the merged lattice with potentially increased interaction range and distance no less than the minimum distance of the merged codes.

For the proof of Theorem 1, we refer to Methods section, where the required SLS formalism is described in full detail. In this approach, the merged code is effectively treated as a subsystem code and the original codes $\mathcal{G}^A$ and $\mathcal{G}^B$ are recovered by gauge fixing. Note that in the merged code, some stabilizers at the

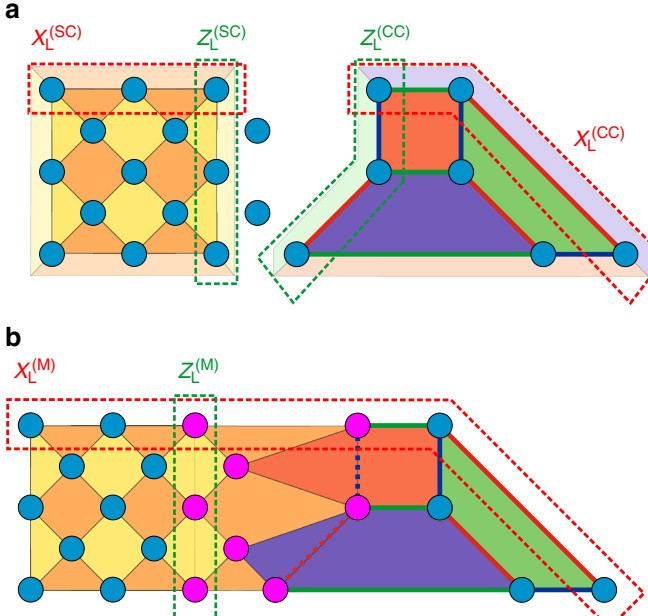

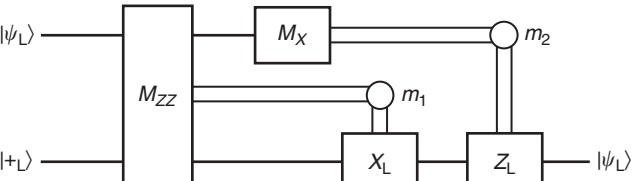

**Fig. 3** Circuit for measurement-based information processing implemented via lattice surgery. The upper and lower single, horizontal lines correspond to the logical qubits encoded in the color and surface codes, respectively. $M_{ZZ}$ and $M_X$ are projective measurements that project onto an eigenstate of $Z_L \otimes Z_L$ and $X_L \otimes \mathbb{1}_L$ respectively. $M_{ZZ}$ is implemented through the merging protocol and code splitting described in the main text. The double lines labeled $m_1$ and $m_2$ represent classical conditioning of the $X_L$ and $Z_L$ gates on the measurement outcomes $m_1$ and $m_2$, respectively

**Fig. 2** Lattice surgery between a surface and a color code. **a** Examples of a surface code (SC) and a color code (CC) with distance 3 defined on lattices with open boundaries. Different types of boundaries are color coded. Qubits are located on vertices depicted in blue. While plaquettes in the surface code correspond to $X$-type (yellow) and $Z$-type (orange) stabilizers, respectively, each plaquette in the color code corresponds to both $X$- and $Z$-type stabilizers. Representative logical Pauli operators $X_L$ and $Z_L$ are depicted by dashed lines and are tensor products of single-qubit $X$ and $Z$ operators, respectively. **b** Merging stabilizers are (orange) plaquette operators acting as $Z$ on qubits along the boundary and ancillas (all shown in fuchsia). Measuring all merging stabilizers projects onto a joint eigenstate of the two logical operators $Z_L^{(SC)}$ and $Z_L^{(CC)}$. While $Z$-type stabilizers remain unaffected, the procedure merges $X$-type stabilizers and logical $X$ operators, respectively, as displayed. In particular, the resulting merged code, labeled by M, encodes only one remaining logical qubit defined by $Z_L^{(M)}$ and $X_L^{(M)}$. Note that the geometry of the underlying codes can be chosen to fit the geometry of a specific computational architecture. Note further that there are many different layouts realizing the same error correction code and this figure merely shows a common variant

boundary are merged while others can generate new gauge qubits. More details on the merged code structure in the most general case and in the case of a specific TSSC can be found in the Methods section. After the merging procedure described in steps (i)–(iv), the two individual codes can be recovered by measuring all ancillas in the $X$-basis. In this case the merged code is split and the two logical subspaces are left in a joint eigenstate of their respective Pauli operators. Here it is important to note that error correction has to be performed on the separate codes in order to retain full fault tolerance[13]. Quantum information can then be teleported from one logical subspace to the other by measurement. This fault-tolerant way of transferring quantum states from one arbitrary TSC (or TSSC) to another provides a simple method that allows exploiting the advantages of different error correction codes. In particular, it can be used to realize an interface between a quantum memory (e.g., based on a surface code) and a quantum processor (using, for example, a color code). We will therefore now demonstrate our protocol for this crucial example of two TSCs.

**Color-to-surface code switching**. The archetypical examples for TSCs are surface codes (SC) and color codes (CC). Surface codes[6]

are defined on 2-face-colorable regular lattices with qubits located on vertices. The two colors are identified with $X$- or $Z$-type generators $S_p^X, S_p^Z \in \mathcal{S}^{(SC)}$, respectively, acting as $S_p^P = \prod_{v \in \mathcal{N}(p)} P_v$, where $\mathcal{N}(p)$ is the set of vertices adjacent to the plaquette $p$ and $P = X, Z$ (Fig. 2a). Horizontal and vertical boundaries cut through plaquettes and can be associated with their respective colors. The lattice is constructed such that opposite boundaries are identified with the same color. The number of logical qubits can be obtained by a simple counting argument. It is given by the difference between the number of physical qubits and the number of independent stabilizers. For the SC, the former is equal to the number of vertices $v$, while the latter equals the number of faces $f$. Since $v - f = 1$, we find that the SC encodes a single logical qubit. Logical operators for SCs are strings connecting boundaries of the same color that are separated by a boundary of a different color. Note that, for any choice of SC, one such logical operator is guaranteed to be aligned with the chosen (e.g., the vertical) boundary[5], but one may not be able to freely choose whether this logical operator is of $Z$- or $X$-type. 2D color codes[8] are defined on 3-face-colorable regular lattices with qubits located on vertices. The generators of the corresponding stabilizer group $\mathcal{S}^{(CC)}$ are pairs of operators $S_p^X$ and $S_p^Z$, i.e., two independent stabilizers per plaquette. The lattice is constructed such that we can use three colors to distinguish three types of boundaries and plaquettes. That is, any two adjacent plaquettes have different colors from each other and from their adjacent boundary (Fig. 2). In the CC, the number of physical qubits is equal to the number of vertices, but there are two independent stabilizers for each face. A quick count then reveals that the CC encodes one ($v - 2f = 1$) logical qubit. Logical operators for CCs are string operators along the lattice connecting three boundaries of different colors. This implies that there exists a string along the boundary of one type that effectively connects all three boundaries.

As an example for the application of our protocol let us consider the situation shown in Fig. 2a, where the outcome of a quantum computation, encoded in an arbitrary logical state $|\psi_L\rangle = \alpha|0\rangle + \beta|1\rangle$ of a color code, is transferred to a quantum memory based on a surface code initialized in the state $|+_L\rangle$. A circuit representation of this task is shown in Fig. 3. Since both initial codes in Fig. 2a have distance 3, we include two ancillas in the state $|+\rangle$, as instructed in step (i). Step (ii) of the protocol can be omitted since the lattices in this example already have the desired structure. As laid out in step (iii) and illustrated in Fig. 2b, one then measures joint $Z$ stabilizers across the boundary, projecting the two surfaces onto a single, merged surface (M) corresponding to an eigenstate of $Z_L^{(SC)} Z_L^{(CC)}$. That is, the merging procedure projects onto a merged code that contains $Z_L^{(SC)} Z_L^{(CC)}$ as a stabilizer. The reduced 2D logical subspace can be represented by $Z_L^{(M)} = Z_L^{(CC)}$ and a merged $X_L$ operator along

both surfaces, e.g., $X_{L}^{(M)} = X^{(SC)} \otimes X^{(CC)}$. Since both color and surface codes are TSCs, the merged code can be understood as a TSC (as opposed to a TSSC) and merging operators coincide with merging stabilizers. Then, as instructed in Fig. 2b, $X$-type stabilizers have to be extended onto ancillas while $Z$ stabilizers remain unaffected. For details on the distinction between TSSCs and TSCs, we refer to the SLS formalism introduced in the Methods. Having obtained this merged code of distance 3, three consecutive rounds of error correction ensure fault tolerance as suggested in step (iv). Note that the merging procedure increases the bias toward $X$-type errors since the minimum weight of logical $X$ operators is increased. However, note that increasing the weight (and interaction range) of $X$ stabilizers on the boundary can cause a temporarily worse error correction performance w.r.t. $Z$ errors for the merged code than for the individual codes.

Irrespective of this we can then proceed by splitting the codes. To this end, we simply measure ancillas in the $X$-basis, while refraining from further measurement of the merging stabilizers. This procedure projects onto the separate surfaces while entangling the logical subspaces. After another three rounds of error correction to preserve fault tolerance, measuring $X_{L}^{(CC)}$ teleports the information according to the circuit displayed in Fig. 3.

## Discussion

We have introduced the method of subsystem lattice surgery (SLS) as a generalization of lattice surgery[13,14] to any pair of 2D topological codes, exploiting their similarities[20]. The applicability of our algorithm to all topological codes such that all topological features are preserved arises from their property to support logical string operators on the boundary[5]. Therefore, the relevance of our algorithm remains unchanged even if other topological codes, such as the subsystem surface code[21] (see also the Methods section) or Bacon-Shor code[22–24], are used as quantum memories or processors. Indeed, our method can even be generalized beyond 2D topological codes at the expense of the 2D layout or topological features (see Methods section for details).

In contrast to the method of code deformation[18,25,26], where a single underlying lattice is smoothly deformed everywhere through multiple rounds of local Clifford transformations, we combine two initially separate lattices through operations within constant distance of their boundaries. To the best of our knowledge, other established methods for code conversion[27–29] have either been applied solely to specific codes, or have not been generalized to subsystem codes.

To highlight the usefulness of incorporating SLS into future quantum computers, let us briefly assess the current contenders for fault-tolerant quantum computation. At present, one of the most promising techniques is SC quantum computation supplemented by magic state injection[30] in order to promote it to a universal quantum computer. However, the overhead on magic state distillation is large[6,7] and it is expected that more effort will be directed towards identifying more resource-efficient techniques. At the same time, other approaches to universal fault-tolerant quantum computation are significantly constrained by no-go theorems that prohibit a universal set of transversal gates in a single stabilizer code[31] and transversal non-Clifford gates in 2D topological codes[32,33]. The former no-go theorem can be circumvented by gauge fixing[11,34] or (subsystem) lattice surgery, and is hence no issue for our approach. The latter no-go theorem can be avoided in a 3D architecture or non-topological codes[35–37]. One potential replacement for magic state distillation is hence the 3D gauge color code[11,38] which successfully sidesteps both no-go theorems. Even though the resource requirement is similar to that of quantum computation with the surface code[38], 3D topological

codes support other useful features such as single-shot error correction[39]. As we show in the Methods section, SLS can also be employed to switch between codes of different dimensions, as well as between topological and non-topological codes. This facilitates the circumvention of both no-go theorems in an elegant fashion. At this point it should also be pointed out that many non-topological codes supporting transversal non-Clifford gates[35–37] have lower resource requirements than comparable 3D topological codes or magic state distillation. However, while all 2D (and some 3D) TSSCs (including the merged code that appears during SLS) with local stabilizers feature error thresholds[19,38], the existence of error thresholds has not been proven for any of the mentioned non-topological codes. That is, in the case of non-topological codes it is not guaranteed that the storage time can be made arbitrarily large by enlarging the code distance. Codes have to be concatenated instead[4].

In any of these cases, quantum computers can only be expected to operate as well as their weakest link. Here, this applies to the error correction code used. The clear advantage of SLS in this context is that the weakest link (i.e., code) may be employed on-demand only and can otherwise be avoided. For instance, in a distributed architecture a code for the implementation of, e.g., a non-Clifford operation can be called only when required. In this scenario, SLS is particularly beneficial since non-Clifford operations could also unfavorably transform errors present in the system[36], while SLS does not carry such errors to other codes. SLS should thus not be seen as a stand-alone contender with other methods of realizing universal fault-tolerant quantum computation, but rather as a facilitator thereof, allowing to selectively combine and exploit the advantages of other methods. We hence expect lattice surgery to play a crucial role in future quantum computers, be it as an element of a quantum bus or for distributed quantum computing[29].

Our findings further motivate experimental efforts to develop architectures that are flexible enough to implement and connect surface and color codes and/or other codes. For instance, ion trap quantum computers[40] may provide a platform soon capable of performing lattice surgery between two distinct codes. In this endeavor, the inherent flexibility attributed to multizone trap arrays[41] may be beneficial. Moreover, topological codes and our approach to code switching are obvious choices for fault-tolerant quantum computation with architectures requiring nearest-neighbor interactions in fixed setups, such as superconducting qubits[42] or nitrogen-vacancy diamond arrays[43]. However, given the local structure of topological codes and the simplicity of our algorithm, the requirements for any architecture are comparatively low.

Despite the inherent fault tolerance of SLS, the codes on which it is performed are nonetheless subject to errors. Further investigations of error thresholds[2,19] for (non-)topological codes and bench-marking[44] are therefore required, in particular with regard to (subsystem) lattice surgery. Finally, our code switching method may find application in the design of adaptive error correction schemes[45].

## Methods

**Subsystem lattice surgery.** In the main text, we have introduced a subsystem lattice surgery (SLS) protocol that is applicable to arbitrary 2D TSSCs, and we have shown how it can be used to teleport quantum information between surface and color codes. In this Methods section, we explain how lattice surgery can be understood entirely within the subsystem stabilizer formalism. To this end, we define SLS by the fault-tolerant mapping $\mathcal{G}^{A} \times \mathcal{G}^{B} \mapsto \mathcal{G}^{A} \times \mathcal{G}^{B} \times \langle P_{L}^{A} \otimes P_{L}^{B} \rangle$, where the merging procedure as described in steps (i)–(iv) and formalized in Theorem 1 is an initialization of an intermediate, merged code $\mathcal{G}^{M}$ and the splitting fixes gauge degrees-of-freedom to obtain $\mathcal{G}^{A} \times \mathcal{G}^{B} \times \langle P_{L}^{A} \otimes P_{L}^{B} \rangle$ from $\mathcal{G}^{M}$.

To see this, we again consider two "standard" TSSCs, $\mathcal{G}^{A}$ and $\mathcal{G}^{B}$. Note that we thereby exclude certain pathological "exotic" codes as will be explained in the proof of Lemma 1. Adopting the previous notation, we assume that the lattices have been

chosen such that the logical operators $P_L^I$ ($I = $ A, B) along their boundaries have support on $N_A$ and $N_B$ qubits, respectively. For ease of presentation we further choose $N_A = N_B = N$. Now, let $\mathcal{G}^I$ with $I = $ A, B be codes characterized by the tuples $[[n_I, k_I, g_I, d_I]]$, where $n_I$ are the numbers of physical qubits, $k_I$ the numbers of logical qubits, $g_I$ the numbers of gauge qubits, and $d_I$ are the distances, respectively. We choose the generating set $\{G_1^I, ..., G_{s_I+2g_I}^I\}$ for code $\mathcal{G}^I$, where $s_I = n_I - k_I - g_I$ is the number of independent stabilizer generators (or the number of stabilizer qubits in accordance with our previous terminology). The lattices are then aligned and prepared as in step (ii) of the algorithm described in the main text, but we omit including ancillas for now. We proceed by defining the merging operators

$$G_i^M = P_{L,i}^A P_{L,i}^B \quad \forall i = 1, ..., N \tag{3}$$

on the merged lattice $\Lambda_M$. From these merging operators, we then collect $\Delta g \leq N - 1$ (where the meaning of this notation becomes apparent in the following section) inequivalent ones (w.r.t. $\mathcal{G}^A \times \mathcal{G}^B \times \langle P_L^A \otimes P_L^B \rangle$) in the set $\mathcal{W} = \{G_i^M\}_{i=1,...,\Delta g}$, and call the group generated by these operators $\mathcal{M}$. We refer to elements of $\mathcal{W}$ as merging generators. This allows us to define the subsystem code

$$\mathcal{G}^M := \langle G_1^A, ..., G_{s_A+2g_A}^A \rangle \times \langle G_1^B, ..., G_{s_B+2g_B}^B \rangle \times \mathcal{M} \times \langle i \rangle. \tag{4}$$

Note that, technically, Eq. (4) specifies a subsystem stabilizer code only if the center of $\mathcal{G}^M$ is non-empty, such that a stabilizer subgroup $\mathcal{S}^M$ can be defined. That this is indeed the case follows from Lemma 2. As we will show next, the code $\mathcal{G}^M$ has the structure of the merged code discussed in Theorem 1. The essential features of this structure can be captured in the following Lemma.

**Lemma 1.** The code $\mathcal{G}^M$ defined in Eq. (4) is a TSSC on the merged lattice $\Lambda_M$ with distance no less than $d_{min} = \min(\{d_A, d_B\})$. In addition, $\mathcal{G}^M$ can be gauge fixed to $\mathcal{G}^A \times \mathcal{G}^B \times \langle P_L^A \otimes P_L^B \rangle$, effectively splitting the code into $\mathcal{G}^A$ and $\mathcal{G}^B$.

*Proof.* First, let us show that $\mathcal{G}^M$ obeys the locality condition for TSSCs. By definition, generators of the separated codes are also generators of the merged code. Their interaction range is increased by at most a constant along the vertical direction due to the relabeling in step (ii). Thereby, the interaction range of merging generators is also kept constant. The generators of the code $\mathcal{G}^M$ are hence all local.

Second, we verify that the distance of the merged code is at least $d_{min}$. Since $\mathcal{G}^A \times \mathcal{G}^B \times \langle P_L^A \otimes P_L^B \rangle$ is a subgroup of $\mathcal{G}^M$, the normalizer $N(\mathcal{G}^M)$ is a subgroup of $N(\mathcal{G}^A \times \mathcal{G}^B \times \langle P_L^A \otimes P_L^B \rangle)$. Then, note that there exist stabilizers $S \in \mathcal{S}^A \times \mathcal{S}^B$ that anticommute with some $G^M \in \mathcal{M}$. If that was not the case, all $G^M|_{Q_I}$ would either be elements of the code $\mathcal{G}^I$ or undetectable errors. The latter option can be ruled out in accordance with the argument in the proof of Lemma 2. The former option, $G^M|_{Q_I} \in \mathcal{G}^I \forall G^M \in \mathcal{M}$, would imply that $P_L^I$ is not a logical operator. In a quantum error correction code, such mutually anticommuting operators act either on the logical subspace or on gauge qubits. However, any stabilizer of the separate codes is also contained in $\mathcal{G}^M$ according to the definition of the merged code in Eq. (4). Now recall that if $\mathcal{G}^M$ is a subsystem stabilizer code, it can be decomposed as $\mathcal{G}^M = \mathcal{L}_G^M \times \mathcal{S}^M \times \langle i \rangle$, as explained in the main text. Therefore, the aforementioned anticommuting stabilizers $S$ must act on gauge qubits and we can hence identify $S$ as belonging to $\mathcal{L}_G^M$. Later, we will further elaborate on the structure of $\mathcal{G}^M$. Since $\mathcal{S}^A \times \mathcal{S}^B$ contains elements that are not in the stabilizer of $\mathcal{G}^M$ but not vice versa, $\mathcal{S}^M$ must be a subgroup of $\mathcal{S}^A \times \mathcal{S}^B \times \langle P_L^A \otimes P_L^B \rangle$. Consequently, any equivalence class of $N(\mathcal{G}^M)/\mathcal{S}^M$ is contained in an equivalence class of $N(\mathcal{G}^A \times \mathcal{G}^B \times \langle P_L^A \otimes P_L^B \rangle)/(\mathcal{S}^A \times \mathcal{S}^B \times \langle P_L^A \otimes P_L^B \rangle)$.

The quotient group $N(\mathcal{G})/\mathcal{S}$ defines so-called bare logical operators that do not act on gauge qubits. Generally, $N(\mathcal{G})/\langle i \rangle \mathcal{S}$ contains all information about the logical operators but not about all undetectable errors. At the same time, undetectable errors are related to bare logical operators, since $N(\mathcal{S})/\mathcal{G} \simeq N(\mathcal{G})/\langle i \rangle \mathcal{S}$ for any subsystem stabilizer code $\mathcal{G}^{19}$. To conclude then that the distance of $\mathcal{G}^M$ is $d_{min}$, we have to verify that deformations (i.e., multiplications) of any non-trivial, logical operators $P \in N(\mathcal{G}^M)/\mathcal{S}^M$ by operators in $\mathcal{G}^M$ yield operators of weight at least $d_{min}$. Since any such $P$ is contained in $\mathcal{L}^A \times \mathcal{L}^B$ this is true for deformations by operators in $\mathcal{G}^A \times \mathcal{G}^B$. But we still need to confirm this for operators in $\mathcal{G}^M - (\mathcal{G}^A \times \mathcal{G}^B)$. Specifically, we have to consider deformations under $\langle G_i^M \rangle_{i=1,...,N}$. In principle, these merging operators can reduce the weight of operators $P$ below $d_{min}$. However, this can only happen under rather exotic circumstances. That is, only for logical operators of the form $P = P^A \otimes P^B$, where $P^I \in \mathcal{L}^I$ ($I = $ A, B) and $P^I \sim P_L^I$ acts as $P_L^I$ on a region $K_I \subset Q_I$ such that for the complementary regions $\overline{K}_I$ (with $\overline{K}_I \cup K_I$ being the set of all qubits on lattice $\Lambda_I$ and $\overline{K}_I \cap K_I = K_I - K_{\overline{I}}$) the weights of these operators satisfy $0 < w(P^A|_{\overline{K}_A}) + w(P^B|_{\overline{K}_B}) < d_{min}$ independently of the system size. Here we make use of the fact that $Q_A$ and $Q_B$ have been labeled equally (cf. main text) and write with a slight abuse of notation $\overline{K}_I \cap K_I = K_I - K_{\overline{I}}$ (where $\overline{I} = $ B if $I = $ A and vice versa). In the scenario described above, one could define the logical operator $P^A \otimes P^B \otimes \prod_{i \in K_A \cap K_B} G_i^M$ with weight $0 < w(P^A|_{\overline{K}_A}) + w(P^B|_{\overline{K}_B}) < d_{min}$. At the same time, this also constricts $P_L^I$ since $P^I$ must be such that $w(P^I \otimes P_L^I) \geq d_{min}$. Since this describes extremely

restricted cases that (to the best of our knowledge) do not feature in any practical scenario, we will exclude codes with such properties from our construction.

Note that we can also exclude cases where $w(P^A|_{\overline{K}_A}) + w(P^B|_{\overline{K}_B}) = 0$. In such cases at least two logical operators $P^A, P^B \in \mathcal{L}^A \times \mathcal{L}^B$ with $\text{supp}(P^I) \subset \text{supp}(P_L^I)$ ($I = $ A, B) exist. However, we are only interested in projecting onto the joint eigenstate of any two logical operators $P_L^A$ and $P_L^B$ along the boundary. Therefore, we can always exclude the aforementioned case by choosing $P_L^A$ and $P_L^B$ such that there does not exist a logical operator that has support on a subset of $Q_I$. In the case of TSSC with non-local stabilizers, such logical operators may have support on disconnected regions of a lattice$^{19}$. Then, these regions have to be connected using local generators to obtain proper string operators. Then, any non-trivial logical operator of the merged code has support on at least $d_{min}$ qubits. The distance of the merged code is hence at least $d_{min}$.

Third, the merged code is scalable since the separate codes are scalable. Thus, we can conclude that $\mathcal{G}^M$ is indeed a TSSC. At last, note that fixing $\mathcal{S}^M$ to $\mathcal{S}^A \times \mathcal{S}^B$ through a measurement of stabilizers $S \in \mathcal{S}^A \times \mathcal{S}^B$ anticommuting with at least one merging generator, the separate codes can be recovered while retaining the eigenstate of $P_L^A \otimes P_L^B$. This gauge fixing has to be accompanied by error correction to ensure full fault tolerance. Q.E.D.

To finally prove Theorem 1, we will now explain the connection between the merged code $\mathcal{G}^M$ defined in Eq. (4) and the framework discussed in the main text. The merging procedure described in steps (ii)–(iv) can be understood as an initialization of the merged code. Specifically, measuring merging generators as in Eq. (3) (or similarly, Eq. (2)) initializes $\mathcal{G}^M$ from the prepared codes $\mathcal{G}^A \times \mathcal{G}^B$. The parity of all random measurement outcomes yields the parity of $P_L^A \otimes P_L^B$ and error correction has to be performed thereafter in order to ensure correctness (see step (iv)). Fault tolerance is guaranteed since merging generators are constant-weight and errors can only propagate to a constant number of physical qubits. Since we have established in Lemma 1 that $\mathcal{G}^M$ is indeed a TSSC with distance at least $d_{min}$, this concludes the proof of Theorem 1.

Altogether, by proving Theorem 1 and Lemma 1, we showed that SLS is fault-tolerant and well-defined through a merging, i.e., a mapping $\mathcal{G}^A \times \mathcal{G}^B \mapsto \mathcal{G}^M$, followed by a splitting, i.e., a mapping $\mathcal{G}^M \mapsto \mathcal{G}^A \times \mathcal{G}^B \times \langle P_L^A \otimes P_L^B \rangle$.

Concluding this Methods section, we return to the discussion of the ancillas added during step (i) of the preparation. As already discussed in the main text, ancillas can be understood as additional stabilizer qubits included into the initial codes. In the merged code these become gauge qubits. The usefulness of ancillas arises when one considers TSCs as it is the case for the example discussed in the Results section. In that case the merged code can be understood as a TSC with stabilizers merged across the boundary. Here, introducing ancillas means that stabilizers of one surface are instead merged with stabilizers of ancillas, allowing for the possibility of reducing the interaction range of the merged code.

**Merged code structure**. In this section, we provide a careful analysis of the merged code. In particular, we will discuss the structure of the subgroups $\mathcal{S}^M$ and $\mathcal{L}_G^M$. Here we claim that $\mathcal{G}^M$ is a $[[n_A + n_B, k_A + k_B - 1, g_A + g_B + \Delta g, d_M]]$ code with distance $d_M \geq d_{min}$. That is, the distance $d_M$ merged code has $n_A + n_B$ physical qubits, $k_A + k_B$ logical qubits, and $g_A + g_B + \Delta g$ gauge qubits, where $\Delta g$ is the number of merging generators (see Eq. (3)). To see this, we have to understand the structure of $\mathcal{S}^M$ and $\mathcal{G}_G^M$ in comparison with the separate codes $\mathcal{G}^A \times \mathcal{G}^B$.

First, note that we can define $\Delta g$ new gauge qubits via their respective Pauli operators $\langle G_i^M, S_i \rangle_{i=1,...,\Delta g}$ where $S_i \in S^A \times S^B \forall i$ such that $[G_i^M, S_j] = 0$ for $i \neq j$. That is, there exist $\Delta g$ independent stabilizers which are now included as gauge operators together with the same number of merging generators. As we will see, the existence of such stabilizers is guaranteed by the choice of logical operators $P_L^A$ and $P_L^B$ as explained in the proof of Lemma 1.

**Lemma 2.** For the operators $P_L^A$ and $P_L^B$ chosen above there exists a stabilizer $S_i \in \mathcal{S}^A \times \mathcal{S}^B$ for any merging generator $G_i^M \in \mathcal{W}$ such that $\{G_i^M, S_i\} = 0$ and $[G_j^M, S_i] = 0 \forall G_j^M \in \mathcal{W} - \{G_i^M\}$.

*Proof.* Suppose such a stabilizer does not exist. Then, there exists a $G \in \mathcal{M}$ that yields the same error syndrome as $G_i^M$, i.e., $\langle \psi | G_i^M G | \psi \rangle = 0$ for any codeword $|\psi\rangle \in \mathcal{C}^A \times \mathcal{C}^B$. Since $G_i^M G \notin \mathcal{G}^A \times \mathcal{G}^B$, it is a non-trivial logical operator in $\mathcal{L}^A \times \mathcal{L}^B$. Combining this with the fact that $\mathcal{W} \subset \{G_j^M\}_{j=1,...,N}$, the existence of the logical operator $G_i^M G$ is not compatible with our construction of $P_L^I$ as discussed in the proof of Lemma 1. That is, there exist no logical operators $P^I \in \mathcal{L}^I$ with $\text{supp}(P^I) \subset \text{supp}(P_L^I)$ ($I = $ A, B) for our choice of $P_L^I$. Q.E.D.

We have hence defined $\Delta g$ new gauge qubits by $\mathcal{W}$ and the same number of independent stabilizers. As another consequence of Lemma 2, we can also define an additional stabilizer equivalent to $P_L^A \otimes P_L^B$. That is, the merged code has $s_A + s_B - \Delta g + 1$ independent stabilizers and thus, a necessarily non-empty center (but reduced logical subspace).

To complete our analysis of the merged code, we have to verify that $\mathcal{G}^M$ still contains $g_A + g_B$ gauge qubits besides new ones. Therefore, note that there might exist pairs of gauge operators $g, \tilde{g} \in \mathcal{L}_G^A \times \mathcal{L}_G^B$ generating a gauge qubit, for which $\{g, G_j^M\} = 0$ with $G_j^M \in \mathcal{W} \forall j \in J$ where $J \subseteq \{1, ..., \Delta g\}$. While the associated gauge qubit in $\mathcal{L}_G^A \times \mathcal{L}_G^B$ was defined by $\langle \tilde{g}, g \rangle$, in the merged code it is redefined to be generated by $\langle \tilde{g}, g \otimes \prod_{j \in J} S_j \rangle$. Here, $S_j \in \mathcal{L}_G^M$ is associated with one of the $\Delta g$ new gauge qubits introduced above. W.l.o.g. we have assumed that $[\tilde{g}, G_j^M] = 0 \forall j = 1$,

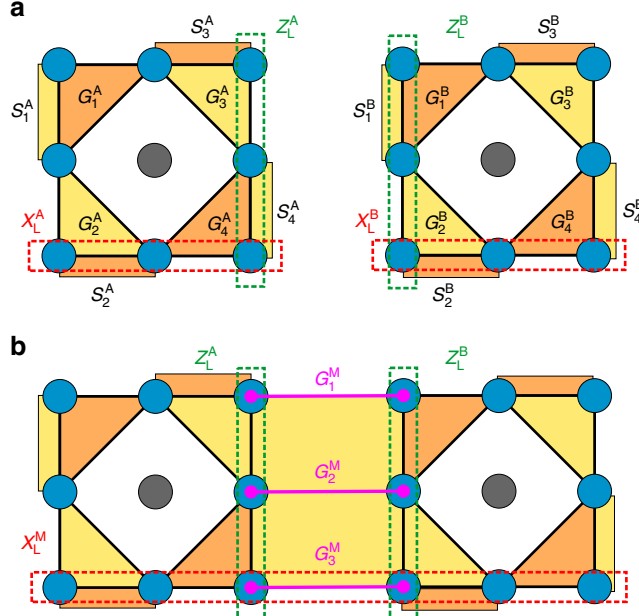

**Fig. 4** Subsystem lattice surgery between two subsystem surface codes. **a** Example of two subsystem surface codes with distance 3 labeled by A, B, respectively. Qubits are located on vertices depicted in blue, while the central gray vertices are unoccupied. Plaquette generators $G_i^I$ and $S_i^I$ ($I =$ A, B) are color coded with $X$-type and $Z$-type operators drawn in yellow and orange, respectively. Representative logical Pauli operators $X_L^I$ and $Z_L^I$ are depicted by dashed lines and are tensor products of single-qubit $X$ and $Z$ operators, respectively. **b** The merged code, labeled by M, with additional merging generators $G_i^M$ colored in fuchsia. Some $X$-type operators in $\mathcal{S}^M$ and $\mathcal{L}_G^M$ are merged across the boundary in accordance with Eqs. (7) and (8). Including merging generators to the codes reduces the dimension of the logical subspace by one

..., $\Delta g$. Otherwise, we would simply have to apply the same argument to $\tilde{g} \in \mathcal{L}_G^A \times \mathcal{L}_G^B$ with $\{\tilde{g}, G_j^M\} = 0$ for some $j \in J' \subseteq \{1, ..., \Delta g\}$.

One can see that this code is indeed generated as indicated in Eq. (4). Hence, the distinction to the separated codes is that some stabilizers now act on gauge qubits which can be reversed by gauge fixing. Interestingly, applying this formalism to two Bacon-Shor codes[22] on regular square lattices $[1, L]^2$, yields a merged, asymmetric Bacon-Shor code[23] on a $[1, 2L] \times [1, L]$ lattice.

**Subsystem surface code lattice surgery**. Here, we exemplify SLS by means of the subsystem surface code (SSC)[21]. Therefore, consider two such codes $\mathcal{G}^A$ and $\mathcal{G}^B$ defined on regular square lattices $[1, L]^2$ with qubits located on vertices. In Fig. 4a, we chose the smallest SSC with distance $d = L = 3$ defining a unit cell where the central vertex is unoccupied. The codes are generated by triangle and boundary operators, labeled $G_i^I$ and $S_i^I$, respectively, with $i = 1, ..., 4$ and $I =$ A, B (Fig. 4a). Triangle operators $\{G_i^I\}_{i=1,...,4}$ act on qubits adjacent to a triangular plaquette $p$ as $G_i^I = \prod_{v \in \mathcal{N}(p)} X_v$ for $i = 2, 3$ and $G_i^I = \prod_{v \in \mathcal{N}(p)} Z_v$ for $i = 1, 4$. Boundary operators $\{S_i^I\}_{i=1,...,4}$ are weight-two Pauli operators acting on every other pair of boundary qubits as either $X$-type or $Z$-type operators for $i = 1, 4$ and $i = 2, 3$, respectively. The stabilizer group is defined as

$$\mathcal{S}^I = \langle S_1^I, ..., S_4^I, G_1^I G_4^I, G_2^I G_3^I \rangle. \tag{5}$$

Henceforth, we write $A \propto B$ whenever we identify a group $A$ with a generating set $B$ up to phases. A single gauge qubit is defined up to irrelevant phases by

$$\mathcal{L}_G^I \propto \langle G_1^I, G_3^I \rangle. \tag{6}$$

Logical operators $X_L^I$ and $Z_L^I$ are tensor products of single-qubit Pauli $X$- and $Z$ operators, respectively, connecting two opposite boundaries as shown in Fig. 4a.

Now we initialize a merged code by measuring merging generators $\{G_i^M\}_{i=1,2,3}$ as depicted in Fig. 4b. The merged code is left with 5 stabilizer qubits,

$$\mathcal{S}^M = \langle S_1^A, S_4^B, G_2^A G_3^A S_1^B, S_4^A G_2^B G_3^B, Z_L^A Z_L^B \rangle, \tag{7}$$

where we only kept stabilizers in $\mathcal{S}^A \times \mathcal{S}^B$ that commute with the merging

generators. Consequently, we can identify 2 additional gauge qubits, i.e., 4 in total,

$$\mathcal{L}_G^M \propto \langle G_1^A, G_3^A S_1^B; G_1^B, G_3^B; G_1^M, S_1^A; G_3^M, S_4^A \rangle. \tag{8}$$

Here we included a semicolon between generators of independent gauge qubits and neglected additional phases. Evidently, this code has the structure predicted in the preceding Methods section and is indeed a TSSC with distance 3.

**Beyond 2D topological codes**. Finally, let us discuss the applicability of SLS to higher-dimensional topological codes as well as to non-topological codes. As we shall argue, all of the above holds true in these cases. In short, this is because the essential feature of SLS is to project onto a joint eigenstate of two logical operators by measuring generators of a merged code. Such a merged code can always be formally defined, irrespective of whether or not the initial codes are topological.

Let us first consider topological codes in more than two dimensions. For instance, 3D topological codes (as opposed to 2D topological codes) can support membrane-like logical operators on their 2D boundary[46,47]. Similarly, it can be expected that most topological codes in $D$ dimensions can support logical operators that are associated with $(D-1)$-dimensional extended objects on the boundary[46]. With this assumption, it is straightforward to show that SLS can be applied in any dimension. This leaves us with the question of whether SLS can also be used to switch between dimensions. This is indeed the case, since, interestingly, 3D topological codes also support string-like logical operators[33] which can be used to teleport information from a 2D topological code to a higher-dimensional one via SLS. In fact, at the expense of its 2D layout, a 2D code can even be wrapped along the surface of a 3D code to perform SLS between a string- and a membrane-like logical operator while preserving a 3D notion of locality.

Finally, let us consider the effects of relaxing the constraint of demanding topological features for the quantum error correction codes under scrutiny. For such codes there exists no notion of locality or scalability. Therefore, an underlying lattice of physical qubits cannot be defined in a meaningful way. However, when revising the arguments from Lemmas 1 and 2 in the spirit of non-topological codes, it turns out that one does not require any topological features to prove that the distance of the merged code is the minimum distance of the initial codes. Neither is it necessary to require locality or scalability to show that the merged code can be gauge-fixed to the original codes. The merged code as specified in Eq. (4) is always well-defined algebraically even without topological features. That is, at the expense of locality and/or scalability, Lemmas 1 and 2 hold and we can use SLS to switch between topological and non-topological codes. Nevertheless, one can expect that scalability in particular is a feature that is desirable for any code. Therefore, let us note that the merged code is scalable if the separate codes are scalable even if it is not topological otherwise. As an example of codes that are scalable but not topological consider the doubled codes[36] which are also amenable for SLS.

**Data availability**. Data sharing is not applicable to this article as no data sets were generated or analyzed during the current study.

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

## Acknowledgements

We are grateful to Rainer Blatt, Nicolas Delfosse, Vedran Dunjko, Alexander Erhard, Esteban A. Martinez, and Philipp Schindler for valuable discussions and comments. We acknowledge support from the Austrian Science Fund (FWF) through the DK-ALM: W1259-N27, the SFB FoQuS: F4012, the START project Y879-N27, and the Templeton World Charity Foundation Grant No. TWCF0078/AB46.

## Author contributions

H.P.N. has devised the code switching protocol, proven Theorem 1 and composed the manuscript. N.F. has contributed to the conceptual development of the project and helped writing the manuscript. H.J.B. has supervised the project. All authors have discussed the results.

## Additional information

**Competing interests:** The authors declare no competing financial interests.

