## [Peer Review File · Nature Communications]

Reviewers' comments:

Reviewer #1 (Remarks to the Author):

The authors have shown that an obvious approach to lattice surgery between a color code qubit and a surface code qubit works for transferring data between the two codes.

My recommendation for this paper is Minor Revision. The content is clear and the paper a pleasure to read. Although the approach taken seems obvious, as far as I am aware no one has tested this before and I am even somewhat surprised that it actually works. It is a valuable theoretical contribution, and may prove to be practically useful, as well.

I do have one moderately important technical concern and a number of smaller comments.

In the explanation of the color code, X_L^{CC} and Z_L^{CC} are both described as operators on the qubit, but with both X and Z stabilizers on all three plaquettes in the figure, it is not clear where the X/Z asymmetry arises; why is the green boundary the Z operator and the blue boundary the X operator? (Keep in mind that the surface code community is still small, and the color code community smaller still, and increasing the number of people who understand them both is important.)

In conjunction with this, in Fig. 2a, it is not clear how to choose how the merge boundary is constructed, and which plaquettes are expanded and how much. Is it always exactly one qubit added to each plaquette?

I am also a little uncertain about the impact of the merged X_L^M operator having plaquettes with only Z stabilizers (orange) and *both* X and Z stabilizers (red and green). Why is this heterogeneity okay? Why is it okay that the merged 22-qubit structure has 12 Z stabilizers and 9 X stabilizers, rather than the same numbers or close?

One item that will help clear this up is to count stabilizers for the reader. Appendix B goes through some of the argument, but showing here that the surface qubit has 13 qubits - 12 independent stabilizers = 1 logical qubit, and similar counting for the color code and the merged surface, will help.

This can probably be cleared up with a little text and prominent references to App. B, although I am not encouraging a significant expansion here -- there is already a great deal of content in the surrounding text, and it is easy to be overwhelmed. But supporting the intuition of the reader in simple language, and guiding those who haven't spent much time thinking about plaquettes and stabilizers, will help.

An important but small correction:

p. 2: "E is not correctable" -- almost but not quite. More correctly,

the code distance is the number of bit flips between the two closest code words, and hence the smallest weight of errors that will move between two logical states entirely undetected. A distinction between "undetected" and "results in miscorrection" might be appropriate here. For many quantum codes, more than $d/2$ errors results in miscorrection and a logical error. For the surface code and color code, many larger error weights can be effectively correctable, but the decoding of syndromes is difficult.

A few minor comments:

To really justify the broad title, a paragraph on other code conversion schemes and their relevance to architecture might be in order. Byung-Soo Choi, Shota Nagayama, Ashley Stephens, Jonas Anderson, and Charles Hill have all proposed code conversion mechanisms for other codes in recent years in the context of architectures or networks. Is there any relationship between their work and yours? Would you deploy your technique along with one or more of these? Are any of them more general?

I would introduce gauge qubits versus program qubits more explicitly. The formal definition late in Sec. I is correct, but its implications are hard to recognize at that point in the paper. This is only a recommendation, though, you may ignore if you choose.

Fig. 1b: Are some of the dashed green lines misplaced?

Fig. 3 somewhat obscures that the merge reduces the number of logical qubits in the system, and the split increases them. Can this be clarified?

Are TSSCs defined in this paper for the first time, or has someone else investigated them? If the latter, please reference.

What is SLS?

I believe the authors intend the "/" to be "set minus", but that is a backslash (or \setminus in LaTeX). Or have I misinterpreted the notation?

In Fig. 4b, I had trouble finding the "purple". In my print they are magenta at best and arguably closer to pink.

App. B has what appears to be $[[n,k,g,d]]$ notation, but the more common form $[[n,k,d]]$ omits the g term, so explaining this notation will help many readers.

Reviewer #2 (Remarks to the Author):

This is an interesting paper on an important topic, which I think may potentially warrant publication in Nature Communications. However, I think the ramifications and limitations of the core technical results have not been sufficiently described in the manuscript in its present form. A more thorough discussion of this point would both improve the paper and make it more feasible to

decide whether it should be accepted in a highly selective journal such as Nature Communications.

I think the most important question about this work is what it does or does not imply about the prospects for universal fault-tolerant quantum computation. There is a lot of prior work here, and the relationship of this result to it needs to be fleshed out. Currently, the top contender for fault-tolerant quantum computation seems to be surface codes plus magic state distillation. However, this scheme leaves much to be desired. In particular, the magic state distillation involved in implementing T gates has high overhead. One could avoid this if one had a code admitting a universal set of quantum gates all of which can be implemented transversally. Eastin and Knill have a no-go theorem which says that this cannot be achieved by any fixed code. I think this should be cited and discussed. The reference is:

B. Eastin and E. Knill, Restrictions on Transversal Encoded Quantum Gate Sets, Phys. Rev. Lett. 102, 110502, 2009.

However, methods have been proposed to circumvent this theorem. As I understand, one such idea is, roughly, to switch back and forth between codes, and one is called gauge fixing. If I understand correctly, some examples of schemes for circumventing Eastin-Knill which probably should be referenced are:

S. Bravyi and A. Cross, Doubled color codes, arXiv:1509.03239

T. Jochym-O'Connor and S. D. Bartlett, Stacked codes: Universal fault-tolerant quantum computation in a two-dimensional layout, Phys. Rev. A 93, 022323 (2016)

However, it is not so clear that these constructions actually lead to a threshold for fault-tolerant universal quantum computing. In fact, there are at least two more no-go theorems to contend with, which should be cited and discussed:

F. Pastawski and B. Yoshida, Fault-tolerant logical gates in quantum error-correcting codes, Phys. Rev. A 91, 012305, 2015

and

S. Bravyi and R. König, "Classification of topologically protected gates for local stabilizer codes," Phys. Rev. Lett., 110(17), 170503, 2013.

So, some key questions about the present work are:

*Are the authors claiming here that their scheme should yield a fault-tolerance threshold?

-if so, how does one thread the needle between the various no-go theorems? The introduction makes it seem that maybe the key point is that the gauge color codes of Bombin that the present work can interface surface codes to, are in 3D, not 2D, and this allows them to avoid the various no-go theorems about 2D and implement a universal set of gates transversally by gauge fixing. However, the main body of the paper, namely section II, discusses only the 2D case. So, it seems like some additional clarification is called for regarding the relationship between the motivations listed in the introduction and the main technical results obtained in section II.

*How does the scheme here compare against the leading candidate, namely surface codes plus magic state distillation?

Response to Referee #1

We would like to thank referee #1 for his or her careful review, helpful comments and recommendation for publication in Nature Communications. Here, we would like to give short, directed answers to each of the referee's remarks (shown indented and in Italic font).

"The authors have shown that an obvious approach to lattice surgery between a color code qubit and a surface code qubit works for transferring data between the two codes.

My recommendation for this paper is Minor Revision. The content is clear and the paper a pleasure to read. Although the approach taken seems obvious, as far as I am aware no one has tested this before and I am even somewhat surprised that it actually works. It is a valuable theoretical contribution, and may prove to be practically useful, as well."

We thank the reviewer for this assessment of our work. In particular, we are glad that our work was found to be a "valuable theoretical contribution". Because of the practical usefulness of our results, we feel that it is of importance that our manuscript is understandable for a wide variety of physicists, which is why we are particularly grateful that the referee has taken the time to also provide comments that help to improve readability.

"I do have one moderately important technical concern and a number of smaller comments.

In the explanation of the color code, $X_{L^{CC}}$ and $Z_{L^{CC}}$ are both described as operators on the qubit, but with both X and Z stabilizers on all three plaquettes in the figure, it is not clear where the X/Z asymmetry arises; why is the green boundary the Z operator and the blue boundary the X operator? (Keep in mind that the surface code community is still small, and the color code community smaller still, and increasing the number of people who understand them both is important.)"

We agree with the referee's observation that some readers may not be familiar with both codes. We have edited the introduction to surface and color codes accordingly, expanding a bit more (within reason) on their basic features whilst trying not to overwhelm the reader with too many details. The apparent asymmetry is by choice only. The X stabilizer of the color code in Fig. 2(a) could have equivalently be chosen on the same qubits as the Z stabilizer. However, we have chosen to display the equivalent choice of the X stabilizer to be placed along the upper boundary to mirror the situation encountered for the surface code on the left-hand side.

"In conjunction with this, in Fig. 2a, it is not clear how to choose how the merge boundary is constructed, and which plaquettes are expanded and how much. Is it always exactly one qubit added to each plaquette?"

In the example of Fig. 2 it is true that each plaquette is expanded by one qubit. However, not all operators associated with a plaquette are expanded. Really, it is the X -type stabilizers which are extended onto ancillas while Z stabilizers remain unaffected. In general, the boundary of the merge is fully determined by the lattice preparation and merging that we explain in Section II.A of the manuscript, in particular, by Eq. (2). If the codes under consideration are TSC, then we can merge stabilizers across the boundary as in the example and obtain a merged code that is a TSC. In the most general case, the merged code is a TSSC and former stabilizers along the boundary act on new gauge qubits. However, the stabilizer group of the merged code still contains those stabilizers that are merged across the boundary.

*"I am also a little uncertain about the impact of the merged X_L^M operator having plaquettes with only Z stabilizers (orange) and *both* X and Z stabilizers (red and green). Why is this heterogeneity okay? Why is it okay that the merged 22-qubit structure has 12 Z stabilizers and 9 X stabilizers, rather than the same numbers or close?"*

One item that will help clear this up is to count stabilizers for the reader. Appendix B goes through some of the argument, but showing here that the surface qubit has 13 qubits - 12 independent stabilizers = 1 logical qubit, and similar counting for the color code and the merged surface, will help."

This is of course an interesting observation. The origin of this phenomenon is in the choice of logical subspaces that are to be entangled. In our example, we choose to project onto a joint eigenstate of the logical Z-operators. In order to achieve this, we have to add Z-type plaquette operators while X-type stabilizers are only expanded. This leads to the different number of X and Z stabilizers of the merged code (and hence, a possibly biased performance). Still only $1=n-s$ qubit is encoded, since there are $s=12+9$ stabilizers, and $n=22$ qubits. As suggested, we have added counting arguments to the extended explanations of the surface and color codes to help shed some light on this matter.

"This can probably be cleared up with a little text and prominent references to App. B, although I am not encouraging a significant expansion here -- there is already a great deal of content in the surrounding text, and it is easy to be overwhelmed. But supporting the intuition of the reader in simple language, and guiding those who haven't spent much time thinking about plaquettes and stabilizers, will help."

We completely agree and hope that the expanded explanations we have included help to clarify whilst keeping the length of the text within reasonable limits.

"p. 2: "E is not correctable" -- almost but not quite. More correctly, the code distance is the number of bit flips between the two closest code words, and hence the smallest weight of errors that will move between two logical states entirely undetected. A distinction between "undetected" and "results in miscorrection" might be appropriate here. For many quantum codes, more than $d/2$ errors results in miscorrection and a logical error. For the surface code and color code, many larger error weights can be effectively correctable, but the decoding of syndromes is difficult."

This is indeed a crucial point. We are thankful to the referee for pointing out this distinction. We have edited the corresponding paragraph accordingly to distinguish between detectable and correctable errors.

"A few minor comments:

To really justify the broad title, a paragraph on other code conversion schemes and their relevance to architecture might be in order. Byung-Soo Choi, Shota Nagayama, Ashley Stephens, Jonas Anderson, and Charles Hill have all proposed code conversion mechanisms for other codes in recent years in the context of architectures or networks. Is there any relationship between their work and yours? Would you deploy your technique along with one or more of these? Are any of them more general?"

To our knowledge, the best known code conversion scheme to date is code deformation, but we agree that also other schemes are of relevance here. We have hence extended the discussion accordingly and included a number of references as suggested by the referee. The most pronounced difference between the method of SLS and others, is that SLS uses teleportation to switch between

different codes, while other methods change the underlying code by Clifford operations. Hence, SLS could be particularly useful for architectures with a fixed layout.

"I would introduce gauge qubits versus program qubits more explicitly. The formal definition late in Sec. I is correct, but its implications are hard to recognize at that point in the paper. This is only a recommendation, though, you may ignore if you choose."

This is indeed a good point to emphasize. In the spirit of understandability, we have provided a more intuitive explanation of what gauge qubits are.

"Fig. 1b: Are some of the dashed green lines misplaced?"

The green lines should connect blue dots in the order they appear in the red string. As far as we can tell they are not misplaced.

"Fig. 3 somewhat obscures that the merge reduces the number of logical qubits in the system, and the split increases them. Can this be clarified?"

We thank the referee for pointing out this potential source of confusion. Both the merge and the split individually are non-unitary operations. During the merging procedure, we initialize the merged code. This code contains $Z_L^{(SC)} \otimes Z^{(CC)}_L$ as a stabilizer. *Since we increase the number of stabilizers, the number of logical qubits is decreased.* We have emphasized this more strongly in the revised manuscript.

"Are TSSCs defined in this paper for the first time, or has someone else investigated them? If the latter, please reference."

As far as we know, they were first discussed in Ref. [19]. Although this reference had been cited in the previous manuscript, we have added the reference earlier on in the revised manuscript for emphasis.

"What is SLS?"

This acronym refers to Subsystem Lattice Surgery. Since this acronym is used in several sections, we have added explanations of the acronym in the Discussion and at the beginning of the Methods in addition to the original introduction of the abbreviation in the Introduction to improve the readability of the manuscript.

"I believe the authors intend the "/" to be "set minus", but that is a backslash (or \setminus in LaTeX). Or have I misinterpreted the notation?"

In fact, we now emphasize that it is supposed to be "/" since the group of logical operators is a quotient group. That is, equivalent logical operators belong to the same class and "equivalence" is given by multiplication of stabilizers (or gauge operators).

"In Fig. 4b, I had trouble finding the "purple". In my print they are magenta at best and arguably closer to pink."

Thank you for pointing this out. As far as we could determine, the correct name for the color used in the figure is "fuchsia", which we have hence used in the revised manuscript.

“App. B has what appears to be $[[n,k,g,d]]$ notation, but the more common form $[[n,k,d]]$ omits the g term, so explaining this notation will help many readers.”

We have added some text to point this out.

Response to Referee #2

We would also like to thank referee #2 for his or her careful review and insightful comments. Here, we would now like to give short, directed answers to the referee’s remarks (shown indented and in Italic font).

“This is an interesting paper on an important topic, which I think may potentially warrant publication in Nature Communications. However, I think the ramifications and limitations of the core technical results have not been sufficiently described in the manuscript in its present form. A more thorough discussion of this point would both improve the paper and make it more feasible to decide whether it should be accepted in a highly selective journal such as Nature Communications.”

We thank the reviewer for helping us to understand that we had not sufficiently explained the ramifications and broad applicability of our work. Indeed, the reviewer’s comments have stimulated further investigation and lead to new results that we hope have further improved the manuscript. In response to the queries and suggestions, we have expanded the Discussion section to elaborate on the current state-of-the-art contenders for universal fault-tolerant quantum computation and the various no-go theorems restricting them. In particular, we have made an effort to provide a better comparison of subsystem lattice surgery (SLS) to surface code quantum computation, and we have explored the broader applicability of our results to non-topological and higher-dimensional codes, which we discuss in an entirely new section in the Methods. Along with these changes, we have included a number of relevant new references, as suggested by the reviewer.

“I think the most important question about this work is what it does or does not imply about the prospects for universal fault-tolerant quantum computation. There is a lot of prior work here, and the relationship of this result to it needs to be fleshed out. Currently, the top contender for fault-tolerant quantum computation seems to be surface codes plus magic state distillation. However, this scheme leaves much to be desired. In particular, the magic state distillation involved in implementing T gates has high overhead. One could avoid this if one had a code admitting a universal set of quantum gates all of which can be implemented transversally. Eastin and Knill have a no-go theorem which says that this cannot be achieved by any fixed code. I think this should be cited and discussed. The reference is:

B. Eastin and E. Knill, Restrictions on Transversal Encoded Quantum Gate Sets, Phys. Rev. Lett. 102, 110502, 2009.

However, methods have been proposed to circumvent this theorem. As I understand, one such idea is, roughly, to switch back and forth between codes, and one is called gauge fixing. If I understand correctly, some examples of schemes for circumventing Eastin-Knill which probably should be referenced are:

S. Bravyi and A. Cross, Doubled color codes, arXiv:1509.03239

T. Jochym-O'Connor and S. D. Bartlett, Stacked codes: Universal fault-tolerant quantum computation in a two-dimensional layout, Phys. Rev. A 93, 022323 (2016)

However, it is not so clear that these constructions actually lead to a threshold for fault-tolerant universal quantum computing. In fact, there are at least two more no-go theorems to contend with, which should be cited and discussed:

F. Pastawski and B. Yoshida, Fault-tolerant logical gates in quantum error-correcting codes, Phys. Rev. A 91, 012305, 2015

and

S. Bravyi and R. König, "Classification of topologically protected gates for local stabilizer codes," Phys. Rev. Lett., 110(17), 170503, 2013."

As we emphasize in the revised Discussion section, our method is extremely broadly applicable and is able to combine advantages of other techniques for universal fault-tolerant quantum computation (FTQC). In this sense, our approach should not necessarily be seen as being in competition with other methods for FTQC but rather as facilitating FTQC by allowing to selectively exploit the strengths (and avoid the weaknesses) of specific codes as required. In particular, this applies to error thresholds, which carry over to our method from the topological codes used, meaning that (subsystem) lattice surgery may adopt proven thresholds from the codes on which it is performed. While, it is clear that any architecture for quantum computation can only be as strong as its weakest link, the clear advantage of our approach in this context is that the weakest link can be used *on-demand* and can otherwise be avoided. For instance, doubled color codes [36] have been shown to support transversal non-Clifford gates, and quite recently, the Bacon-Shor code [22] has also been proven to allow the implementation of a Toffoli gate [24]. Nonetheless, no proven threshold theorems exist for either code but both are amenable to SLS and can be used *on-demand* to implement non-Clifford operation. In other words, once we find a code that supports non-Clifford operations *and* an error-threshold, we can incorporate it in our architecture using SLS. Whether or not such a combination is efficient has to be evaluate through case-by-case studies. However, we have taken the reviewer's comments to heart and have included an extended discussion of the different approaches mentioned above, along with a number of new references, as suggested by the reviewer.

"So, some key questions about the present work are:

Are the authors claiming here that their scheme should yield a fault-tolerance threshold? -if so, how does one thread the needle between the various no-go theorems? The introduction makes it seem that maybe the key point is that the gauge color codes of Bombin that the present work can interface surface codes to, are in 3D, not 2D, and this allows them to avoid the various no-go theorems about 2D and implement a universal set of gates transversally by gauge fixing. However, the main body of the paper, namely section II, discusses only the 2D

case. So, it seems like some additional clarification is called for regarding the relationship between the motivations listed in the introduction and the main technical results obtained in section II.”

We absolutely agree with the referee that these points are of profound importance and should be emphasized. We have amended and significantly expanded our manuscript to elaborate on them. In short, the answers to the questions above are: Yes, in principle our scheme allows for error thresholds, provided one applies our method to codes that permit such thresholds, and: Yes, our method can be generalized to higher dimensions. Let us expand on these statement.

As we argue in the revised Discussion section, our method should not necessarily be seen as a contender for other methods of universal fault-tolerant quantum computation, but as a facilitator thereof by allowing to selectively exploit the strengths (and avoid the weaknesses) of specific codes as required.

In the context of error thresholds this nonetheless means that such thresholds for the merged code can only be expected if the separate codes have thresholds themselves. The existence of a threshold during SLS thus becomes a question of which specific codes are used. For instance, if the separate codes are topological in nature, then the merged code does have an error threshold (see Ref. [19]). However, a 3D or non-topological code would be needed to circumvent the no-go theorem of Ref. [32,33]. Here, non-topological codes are particularly interesting since they potentially have lower resource requirements than 3D codes. Unfortunately, threshold theorems have not been proven for the most relevant candidates [24, 35-37] among non-topological codes. Once such codes with error thresholds have been found, one may study the threshold behaviour of the merged codes on a case-by-case basis. Should such thresholds be proven for efficient non-topological codes, it could hence be desirable to switch between topological and non-topological codes. This is indeed possible, i.e., we have been able to generalize our method beyond 2D topological codes (see Methods Sec. IV). In particular, SLS between topological codes of different dimensions is now also supported. We hope that the extensions to the Discussion now clarify the perspective of SLS in the spirit of the title of our manuscript.

“How does the scheme here compare against the leading candidate, namely surface codes plus magic state distillation?”

This is a question of fundamental importance and we hope to have answered it in the newest version of the Discussion: A quantum computing architecture that supports SLS has the potential to be significantly better than surface codes plus magic state distillation if we employ non-topological codes for the implementation of non-Clifford gates [24,36,37]. Then, we can potentially lower the overhead that comes from a 3D architecture or magic state distillation (which are approximately the same) by using a non-topological code only *on-demand*, as explained also in the response to the previous question. In fact, 3D codes are already arguably an improvement over surface code quantum computation, since the resource requirement is the same while it can support one-shot error correction [39].

List of changes

Besides the manuscript file, we have provided a pdf where all changes w.r.t. the previous version are highlighted in yellow, except for the reference section. There we have reordered some references in accordance with their appearance in the revised manuscript and we have newly included the following references: 24, 27-39, 44.

REVIEWERS' COMMENTS:

Reviewer #1 (Remarks to the Author):

I am satisfied with the revision and the author's response.

I still think that some of the green lines in Fig. 1b are misplaced, please double check. Some of the qubits terminate two green lines.

I do not need to see the paper again, I am happy for it to be published.

Reviewer #2 (Remarks to the Author):

The authors have done a good job of clarifying the implications of their work for fault-tolerant quantum computing. In my opinion they have made a compelling case and the manuscript may be accepted for publication in Nature Communications without further review or revision.

Response to Referee #1

We would like to thank referee #1 for his or her assessment of our revision. In particular, we are delighted that the reviewer recommended publication in Nature Communications.

"I am satisfied with the revision and the author's response.

I still think that some of the green lines in Fig. 1b are misplaced, please double check. Some of the qubits terminate two green lines.

I do not need to see the paper again, I am happy for it to be published."

We agree that the starting and endpoints of the green lines are not clearly visible and it sometimes seems as if two lines terminate at the same qubits. We have therefore replaced some of the straight green lines with bent curves for better visibility.

Response to Referee #2

We would also like to thank referee #2 for his or her assessment of our revision. In particular, we are delighted that the reviewer is now convinced of the relevance of our work.

"The authors have done a good job of clarifying the implications of their work for fault-tolerant quantum computing. In my opinion they have made a compelling case and the manuscript may be accepted for publication in Nature Communications without further review or revision."